**Brief Communication**

# Rapid and sensitive protein complex alignment with Foldseek-Multimer

Woosub Kim [1,2], Milot Mirdita [2], Eli Levy Karin[3], Cameron L. M. Gilchrist[2], Hugo Schweke[4,5], Johannes Söding[6,7], Emmanuel D. Levy [4,5] ✉ & Martin Steinegger [1,2,8,9] ✉

Advances in computational structure prediction will vastly augment the hundreds of thousands of currently available protein complex structures. Translating these into discoveries requires aligning them, which is computationally prohibitive. Foldseek-Multimer computes complex alignments from compatible chain-to-chain alignments, identified by efficiently clustering their superposition vectors. Foldseek-Multimer is 3–4 orders of magnitudes faster than the gold standard, while producing comparable alignments; this allows it to compare billions of complex pairs in 11 h. Foldseek-Multimer is open-source software available at GitHub via https://github.com/steineggerlab/foldseek/, https://search.foldseek.com/search/ and the BFMD database.

The similarity between two protein complexes is reflected in their optimal structural alignment, which also dictates a pairing of their chains. Aligning and comparing quaternary structures is essential for quantifying their structural diversity and identifying structural similarities and changes across different conformations or homologs. Furthermore, it is important to understanding protein function because many proteins operate as complexes[1].

Recently, Foldseek[2] has been developed as a fast structural aligner to detect similarity between two single-chain proteins, expressed using 3Di, a designated alphabet for describing tertiary amino acid interactions. Using Foldseek allows searching for similar single-chain structures in large databases, such as the AFDB[3]. However, because aligning two complexes requires knowing the correct pairing of their chains, Foldseek cannot be used directly to find the alignment between them.

US-align[4] is a structural aligner for various types of molecules, including protein complexes. Its strategy for complex alignment is TM-score maximization. As there is a factorial number of possible assignments of chain pairings, US-align uses a greedy search heuristic for proposing candidate assignments, which are refined by dynamic programming. This heuristic was shown to make US-align up to five

times faster than the state-of-the-art MM-align[5], while producing higher scoring alignments, making US-align the gold standard for pairwise complex alignment.

Aiming to discover pairs of structurally conserved interfaces in large databases, Dey et al.[6] developed QSalign for the detection of similar homomeric complexes. QSalign saves computation time by performing the full pairwise structural alignment only on complex pairs prefiltered based on their sequence similarity, retaining pairs with around 25% sequence identity or more. This speed-up comes at the expense of sensitivity, limiting its ability to discover structurally similar pairs in the twilight zone or below. Despite this speed-up, QSalign still took several months to conduct an all-versus-all search encompassing about 100,000 complexes in the 3DComplex DB V5 (ref. 7) using 100 threads. An alternative approach to reduce computational time during database search was presented by Guzenko et al.[8], who compared the shapes between two complexes through 3D Zernike descriptors, avoiding the need to pair their chains. This approach can query through hundreds of thousands of structures in less than a second. However, it can only discover global matches between molecules of similar shapes, limiting its sensitivity, compared to chain-pairing

[1]Interdisciplinary Program in Bioinformatics, Seoul National University, Seoul, Republic of Korea. [2]School of Biological Sciences, Seoul National University, Seoul, Republic of Korea. [3]ELKMO, Copenhagen, Denmark. [4]Department of Chemical and Structural Biology, Weizmann Institute of Science, Rehovot, Israel. [5]Department of Molecular and Cellular Biology, University of Geneva, Geneva, Switzerland. [6]Quantitative and Computational Biology, Max-Planck Institute for Multidisciplinary Sciences, Göttingen, Germany. [7]Campus Institute Data Science (CIDAS), University of Göttingen, Göttingen, Germany. [8]Institute of Molecular Biology and Genetics, Seoul National University, Seoul, Republic of Korea. [9]Artificial Intelligence Institute, Seoul National University, Seoul, Republic of Korea. ✉e-mail: emmanuel.levy@unige.ch; martin.steinegger@snu.ac.kr

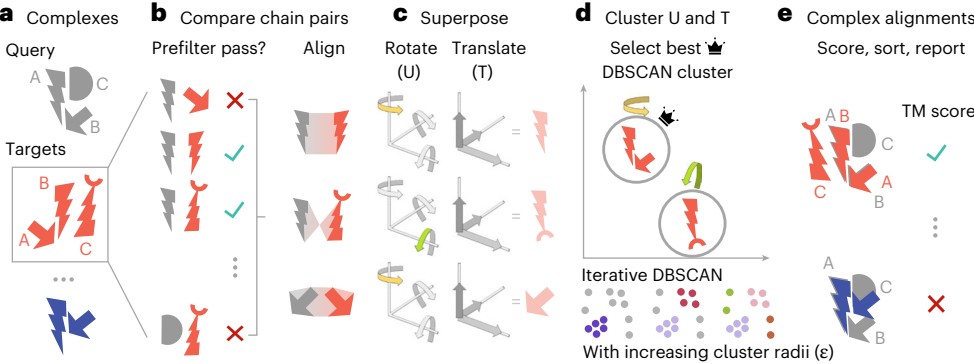

**Fig. 1 | Foldseek-Multimer schematic. a,** Foldseek-Multimer allows fast querying of input complex(es) against a large database, potentially containing millions of targets. **b,** All chains from the query (gray) are compared to those of each target (red). A prefilter allows to quickly reject non-matching chain pairs so the full alignment is only applied to promising complex pairs. **c,** Foldseek-Multimer represents each chain-to-chain alignment as a superposition, described by rotations and translations, required for superposing the target chain onto the query. In this simplified example, two chain-to-chain alignments (top, bottom) are a rotation along one axis (yellow and green highlights), while one (middle) is a rotation along a different axis. **d,** The complex-to-complex alignment is inferred from chain-to-chain alignments as the superpositions of chain pairs in the complex alignment are similar ('Algorithm: overview'). Foldseek-Multimer uses the DBSCAN algorithm iteratively, with increasing radii, to identify superposition clusters and the best-scoring valid cluster for computing the complex alignment (Supplementary Fig. 1). **e,** Based on the best-scoring cluster, the complex TM score is computed across all chain alignments between query and target.

methods, like US-align and QSalign. Furthermore, it is unable to find local matches within chains that do not match globally.

The challenge of sensitively searching large databases is expected to intensify as the computational prediction of protein complexes using tools like AlphaFold-Multimer[9] can now be performed on entire proteomes to systematically predict complexes[10–12] and on sequences from metagenomic samples. This will enrich our databases with a plethora of structures, potentially in the millions, in the coming years.

To address the need for large-scale structural comparisons between complexes, we developed Foldseek-Multimer (Fig. 1). Three factors contribute to its speed: (1) using Foldseek for fast chain-to-chain comparison, (2) describing chain-to-chain alignments as superposition vectors, and using them to identify complex alignments by efficient clustering, and (3) utilizing clustered databases during searches. Through benchmarks, we show that Foldseek-Multimer is: (1) nearly as accurate as US-align, while being orders of magnitude faster, (2) sensitive and suitable for metagenomic studies of complexes with low sequence similarity to others, (3) capable of all-versus-all searches, examining billions of complex pairs in 11 h.

The quality of Foldseek-Multimer's alignments was compared to that of US-align on a benchmark of 931 pairs of protein complexes, known to be structurally similar, using either tool to align them. Foldseek-Multimer was run in two modes, differing in the algorithm used for chain-to-chain alignment: 3Di+AA (Foldseek-MM) or TM-align[13] (Foldseek-MM-TM). Both tools detected the vast majority (>95%) of pairs as similar (US-align: 97.6%, Foldseek-MM-TM: 97.4%, Foldseek-MM: 95.8%), aligning them with a TM score ≥ 0.65, which is a cutoff found to be optimal for detecting structural similarity among complexes[6]. Using either mode, Foldseek-Multimer computed highly correlated TM scores to those of US-align (Fig. 2a and Supplementary Fig. 2) and produced the same chain pairing in >99% of the cases (see 'Data availability').

We measured the runtime of the tools, breaking down the contribution of Foldseek-Multimer's components to its speed. First, given the task of computing 931 pairwise alignments, we observed a speed-up of 1–2 orders of magnitude over US-align (Fig. 2b and Supplementary Fig. 3), reflecting the efficiency of the chain-to-chain alignment (Foldseek-MM) and superposition clustering (Foldseek-MM and Foldseek-MM-TM). The performance of Foldseek-MM-TM thus highlights the key contribution of Foldseek-Multimer's innovative use of superpositions as an alternative to US-align's global alignment. Next, the tools queried each of the 677 complexes in the benchmark

(Methods) against the 3DComplexV7 database[7]. Here, Foldseek-Multimer was 3–4 orders of magnitude faster than US-align (Fig. 2b and Supplementary Fig. 3) due to an additional speed-up by its prefilter.

Recently, Altae-Tran et al.[14] discovered the first CRISPR–Cas type IV-A system with a specified interference mechanism in an environmental sample of *Sulfitobacter* sp. JL08. Intrigued by their finding, we predicted a part of its ribonucleoprotein complex structure using ColabFold-AlphaFold-Multimer[9,15]. The prediction was of acceptable quality (0.564 pTM), and we provided it as a query to Foldseek-Multimer and US-align in a search against the Protein Data Bank (PDB)100 database (Methods). Foldseek-MM and Foldseek-MM-TM demonstrated remarkable efficiency in comparing a query consisting of six chains and spanning 1,843 amino acids against the 426,347 entries of PDB100. These comparisons took only 27 s and 6 min, respectively, on a single core of a server (23 s and 96 s on an 8-core MacBook Pro). By contrast, it took US-align 13 days.

Here, in addition to its fast core algorithm (Fig. 1), Foldseek-Multimer gained further acceleration since PDB100 is a clustered database, allowing it to search against the 343,785 representatives, instead of all entries, and to expand the search only within promising clusters (Methods). Foldseek-MM-TM and US-align scored five entries above 0.65. These entries were the top ranks by Foldseek-MM, scoring above 0.5 but below 0.65 (Fig. 2c, rank indicated by '#'). All five hits were from a recently reported type IV-A system in *Pseudomonas aeruginosa*[16], which belongs to a different class (Gammaproteobacteria) than that of the query (Alphaproteobacteria). When examining the best match, 7xg4, we found that Foldseek-Multimer could identify similarity, despite low sequence similarity (11.1–19.8% sequence identity and 19–33.3% sequence similarity using the BLOSUM62 substitution matrix) between the six subunit pairs of *Sulfitobacter* sp. JL08 and those of 7xg4. This provides further support for the previous identification of the *Sulfitobacter* sp. JL08 system as type IV-A and highlights the potential of Foldseek-Multimer for investigating protein complex structures predicted in distant organisms from environmental samples (Supplementary Fig. 4: prediction quality effect).

Next, we examined Foldseek-Multimer in an all-versus-all setting, using the 3DComplexV7 database[7] as it had been previously analyzed in this setting using QSalign (Methods). QSalign relies on the time-consuming Kpax[17] structural alignment method, which prohibits it from conducting an exhaustive structural search. Thus, it first identified around 58 million pairs, which shared sequence similarity

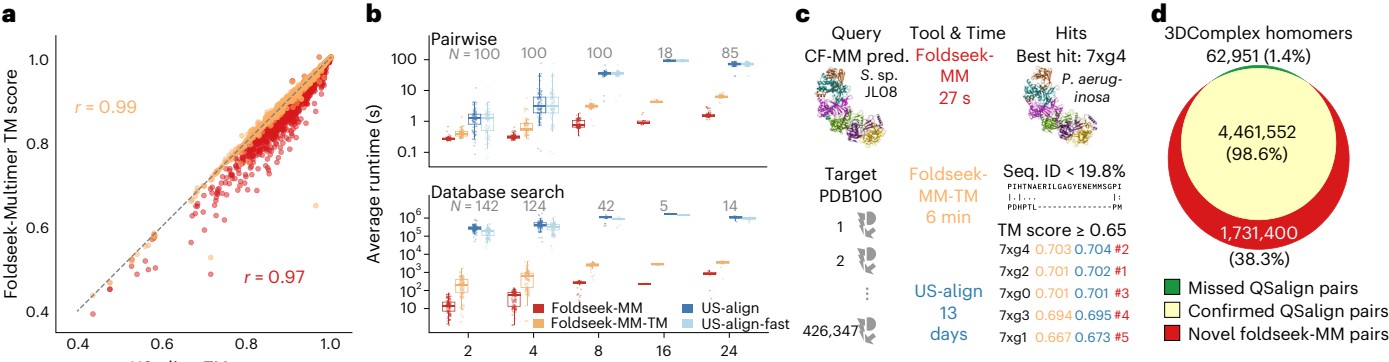

**Fig. 2 | Performance of Foldseek-Multimer. a**, Query-length normalized TM scores (target-normalized: Supplementary Fig. 2) computed for 931 pairs of structurally similar complexes by US-align or Foldseek-Multimer. Both measures correlated highly (Pearson's *r*). **b**, Execution time based on the dataset used for **a**. Complexes were binned by their number of chains; selected bins are shown (for all bins, see Supplementary Fig. 3). Box plots depict quartiles, each point is a complex pair (top) or complex (bottom), sample sizes are indicated as *N*, and whiskers are drawn to the maximum (minimum) point within 1.5 times the interquartile range over (under) the 75th (25th) percentile. Pairwise mode (top): Foldseek-Multimer is 10–100 times faster than US-align due to efficient chain-to-chain alignment and superposition clustering. Database search (bottom):

complexes were queried against 3DComplexV7. Foldseek-Multimer is further accelerated by its prefilter, making it $10^3$–$10^4$ times faster. **c**, An AlphaFold-Multimer prediction of a part of a CRISPR–Cas ribonucleoprotein from an environmental sample (top left) was queried by Foldseek-Multimer and US-align against PDB100. Foldseek-MM-TM identified the same hits as US-align, while being >3,000 times faster. These hits were the top ranks by Foldseek-MM (red) with TM score > 0.5. Non-aligned components of 7xg4 (top right) are set as transparent. **d**, Foldseek-Multimer was run on 57 billion pairs of complexes from 3DComplexV7. It discovered nearly all homomeric pairs previously identified as similar by QSalign, and found an additional 1.7 million homomeric pairs (Supplementary Fig. 5).

and then applied Kpax only to them, detecting around 4.5 million pairs of similar homomers ('QSalign pairs').

Using 128 cores, Foldseek-MM then queried the clustered 3DComplexV7 (Methods) against itself, examining 57 billion pairs in 11 h. Applying the same TM score ≥ 0.65 cutoff as QSalign, Foldseek-MM identified 98.6% of the homomeric pairs previously identified by QSalign and found an additional 1.7 million similar homomeric pairs: 'Foldseek-MM Pairs' (Fig. 2d). We used US-align for evaluating a randomly selected sample of 10% of the Foldseek-MM pairs (Methods). US-align confirmed 98.2% of the sampled pairs and rejected 1.8% (TM score < 0.65). We thus conclude that over 1.6 million of the homomeric pairs are new discoveries by Foldseek-Multimer, owed to its ability to detect similar complex structures below the twilight zone (Supplementary Fig. 5).

In addition to developing a command-line tool, we extended the Foldseek web server to support Foldseek-Multimer and visualize its search results using the NGL viewer library[18]. The web server overlays chain-to-chain assignments by using translucently colored protein surfaces. Users can choose between Foldseek-Multimer's alignment modes, and apply taxonomic filters, restricting the search to specific clades. To accompany the web server with predicted structures, we organized 297,570 multimer predictions from community efforts[10,11,19–21] into a single database (BFMD; Methods). BFMD is available in the web server and for local use.

In conclusion, we presented a strategy for complex-to-complex alignment, which quickly identifies compatible sets of chain-to-chain alignments by their superpositions. Demonstrated here on protein complexes, the Foldseek-Multimer strategy can be extended to other modalities, such as RNA and DNA complex structures, given a way to align their individual subunits. Together, the unprecedented sensitivity and speed offered by Foldseek-Multimer make it an essential tool for investigating complex structures in the AlphaFold era.

## Online content

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

## Methods

### Algorithm: overview

Foldseek-Multimer examines all possible chain-to-chain pairings between the compared complexes, using Foldseek (Fig. 1b). It then uses the fact that a structural alignment between two complexes, Q and T, indicates a superposition: a set of rotations and translations, which minimize the sum of squared distances between their aligned residue pairs[22]. For simplicity, assume Q and T to be two structurally similar dimers, consisting of the chains Q_A, Q_B and T_A, T_B, where Q_A is similar to T_A and Q_B is similar to T_B. The physical meaning of the complex-level structural similarity is that Q_A is positioned and oriented relative to Q_B within Q in the same way that T_A is positioned and oriented relative to T_B within T. Thus, the same superposition, that is, the same set of rotations and translations, would minimize the distance between Q_A and T_A as well as the distance between Q_B and T_B. In other words, all individual chain-to-chain superpositions (for example, the one between Q_A and T_A) are equal to one another and to the complex-to-complex superposition. Therefore, a set of chain-to-chain alignments is compatible and can define a complex-to-complex alignment, only if all chain-to-chain superpositions computed from that set are equal. Therefore, Foldseek-Multimer computes for each chain-to-chain alignment a vector, representing its superposition (Fig. 1c). Next, it uses DBSCAN[23] for clustering these vectors to identify compatible sets of chain-to-chain alignments, which share the same superposition and define valid complex alignments (Fig. 1d and Supplementary Fig. 1). Once complex alignments are identified, Foldseek-Multimer computes their TM score[24] and reports them (Fig. 1e).

### Algorithm: input

Foldseek-Multimer allows for searching one or more query protein complex structures against a target complex structure, a database of complex structures or a database of clustered structures. Structures can be provided in PDB/mmCIF format or as a Foldseek-formatted database. Formatting structures is possible using the `createdb` command.

### Algorithm: chain-to-chain alignments

By utilizing Foldseek, Foldseek-Multimer offers two main modes for chain-to-chain structure comparison. The default mode, 3Di+AA, encodes structures as sequences over a 20-state 3Di alphabet, as fully described by ref. 2. Additionally, chain-to-chain alignments can be computed using TM-align[13], which is a global, albeit slower, alignment method. During database search, a prefilter, which is based on the 3Di+AA mode, allows for fast removal of most chain pairs, continuing to compute chain-to-chain alignments only on promising candidates.

### Algorithm: chain-to-chain superposition vectors

Given a chain-to-chain alignment, Foldseek-Multimer computes the superposition of the target chain onto the query chain, using nine rotations (U) and three translations (T). In preparation for aligning complex structure Q and complex structure T, Foldseek-Multimer creates a matrix with 12 columns, whose rows are the superposition vectors, computed from all chain-to-chain alignments, belonging to Q and T. The mean and the standard deviation (s.d.) of each column are then used to compute the coefficient of variation (CV = s.d./mean) of the column and exclude less-informative columns (CV < 0.1; Supplementary Table 1: the effect of this parameter on Foldseek-Multimer's performance). If the mean value of the column is <1, the s.d. value is used instead of the CV for the exclusion criterion. Finally, the retained columns undergo normalization since they can have different scales. To that end, Foldseek-Multimer subtracts from each column its mean and divides it by its s.d. We denote the resulting reduced and normalized matrix as supQT.

### Algorithm: chain-to-chain clustering

DBSCAN is used iteratively for clustering the rows of supQT as it doesn't require knowing the number of clusters a priori. The stages of this procedure are described below and demonstrated on a small example in Supplementary Fig. 1.

**Initialization.** The Euclidean distances between all row pairs in supQT are computed and the minimum (minDist) initializes the parameter epsilon. The biggest cluster(s) encountered during the procedure are recorded in a candidate list alongside their size (maxClusterSize), which is initialized to 0.

**The DBSCAN iteration.** For each supQT row, all rows within a radius of epsilon from it, are defined as its 'neighbors'. Then, all rows, which have at least one more neighbor (at least two neighbors, including itself) are considered as 'core points' and the rest as 'non-core points'. Next, a core point is selected at random to start the first cluster. All its core-point neighbors are added to the first cluster. Each added core-point neighbor also adds its core-point neighbors and so on, until no more core points can be added to the first cluster. Then, all non-core points, which are neighbors of members of the first cluster, are added to it as well (without adding their neighbors). The second cluster is constructed similarly, operating on the remaining unclustered points.

**Cluster validity and rescuing by nearest neighbors.** During the DBSCAN iteration, after each cluster is computed, Foldseek-Multimer evaluates its validity. If a cluster includes the same chain in multiple chain-to-chain alignments, Foldseek-Multimer attempts to rescue it by selecting a compatible subgroup of points (that is, chain-to-chain alignments) from that cluster. To that end, points are selected for the subgroup in the order of their distance to the core point, which was used to initiate the cluster. Selection for the subgroup is stopped once the process encounters a point that includes a chain, which was already added by a previous point.

At each DBSCAN iteration, valid clusters which are at least as big as maxClusterSize are added to the candidate list. The value of maxClusterSize is updated each time a bigger cluster is encountered and all previously added clusters are removed from the list owing to being smaller.

**Iterativity.** Next, the value of the radius epsilon is increased by a delta of 0.1 (Supplementary Table 1: the effect of this parameter on Foldseek-Multimer's performance) and a new DBSCAN iteration starts, potentially forming new clusters. If all new clusters are smaller than maxClusterSize, the procedure stops. Otherwise, the candidate list and maxClusterSize will be updated with the iteration's clusters and epsilon will increase again, up to a maximal value of the distance between the two furthest points (maxDist).

**Early stop condition.** Let $C_Q$ and $C_T$ be the number of chains in Q and T, respectively. Without loss of generality, assume $C_Q < C_T$. If maxClusterSize is equal $C_Q$, then no bigger valid cluster exists. Since there is a total of $C_Q \times C_T$ chain-to-chain alignments, the number of clusters in the candidate list cannot exceed $C_T$ once maxClusterSize is equal to $C_Q$. Foldseek-Multimer checks these two conditions and avoids unnecessary DBSCAN iterations if they are met.

**Discovered clusters.** At the end of the iterative DBSCAN procedure, the biggest valid clusters are returned. Each of them is equivalent to a set of compatible chain-to-chain alignments with a similar superposition that together define a complex alignment between Q and T.

### Algorithm: TM-score computation

TM scores are computed for the complex alignment derived from each of the valid clusters found for a Q–T complex pair as follows. First, the chains of complex Q are concatenated to each other in some order.

Given the concatenation order of the chains in Q, Foldseek-Multimer concatenates the chains of complex T, in the order of their pairwise matches to the chains of Q, as defined by the cluster. Then, the TM score between the concatenated Q and concatenated T is computed the same way Foldseek computes it for single-chain pairwise alignments, using the C$\alpha$ coordinate vectors of both chains (concatenated chains in this case). Using this computation, all complex alignments a given query complex Q has with a specific target T and with all other target complexes can be ranked and reported by their TM score.

### Algorithm: utilizing clustered databases

To further accelerate Foldseek-Multimer, we aimed to reduce the redundancy in the target database, an approach, which is also adopted by TM-search[25]. To that end, we introduced a new capability to Foldseek, which allows it to efficiently search through clustered databases in MMseqs2 or Foldseek format (for example, PDB100, see below). If the input has M cluster representatives and N cluster members (M < N), Foldseek will first search (prefilter + alignment) against the M representatives, finding candidates below a specific $E$-value threshold (the default value of 10 was used in this study). Extending to promising clusters only, the alignment step will then be carried out on all cluster members of the candidates. Foldseek-Multimer will use the alignment results of all extended clusters for computing superposition matrices and the following procedure steps, as described above.

### The 3DComplex database and QSalign comparisons

For the analyses presented in Fig. 2a,b,d, we downloaded the 3DComplex database version 7 (3DComplexV7 DB; see 'Data availability'). In brief, this database holds 238,965 structures, consisting of 557,146 chains and was created from the 'Biological Units/Assemblies' downloaded from the PDB using the method described previously[7]. Before this study, QSalign[6] had been applied to 3DComplexV7 DB and yielded a list of 57,953,513 compared structural pairs.

### Similar pairs benchmark

**Dataset.** Starting with the list of 57,953,513 QSalign-compared pairs, we selected entries with varying numbers of subunits (from 2 to 24). For each size, the criteria for selection were that the TM score computed by Kpax[17] was greater than 0.8, and that pairs of homomers had less than 80% sequence identity. If more than 100 pairs matched the criteria, only the first 100 were selected, resulting in a total of 931 complex pairs included in the benchmark.

**Runtime evaluation.** Performance was measured on a server with a 1x AMD EPYC 7702P 64-core CPU and 1 TB RAM, using a single core. The queries for the time measurements in Fig. 2b and Supplementary Fig. 3 were the 677 unique complexes associated with the 931 pairs. Owing to its high computational demand, the runtime of US-align on these 677 complexes against 3DcomplexV7 was extrapolated from running against 1,000 randomly sampled 3DcomplexV7 entries. Reporting the average over the number of cases $N_c = 142, 109, 124, 18, 101, 7, 42, 8, 41, 44, 17, 5, 5, 14$ for each number of chains $c = 2, 3, 4, 5, 6, 7, 8, 9, 10, 12, 14, 16, 18, 24$: avg $= \frac{1}{N_c} \sum_{i=1}^{N_c} t(q_i, \text{sample}) \frac{238,965}{1,000}$. Foldseek-Multimer was run against the full database, without extrapolation.

### Environmental CRISPR–Cas

**The PDB100 database.** A version of the PDB, termed PDB100, was used to search for structural homologs of an environmental CRISPR–Cas as well as to measure the runtimes of Foldseek-Multimer and US-align. PDB100 was first introduced by ref. 2, but further developed in this study, as described here. First, PDB, containing the asymmetric unit of 207,937 entries, consisting of 1,047,615 chains, was downloaded in November 2023 (see 'Data availability'). Of these, 11,901 entries were associated with more than one structural model (for example, the NMR experiment

2KOX). In total, 426,347 structural models were associated with the PDB entries. Next, all chains were clustered using Foldseek (parameters: `-c 0.95 --min-seq-id 1.0`), resulting in 343,785 redundancy-reduced representatives. In contrast to van Kempen et al.[2], PDB100 is now a cluster database, which holds the representatives alongside information to associate them to their cluster chains and structural models. PDB100 is updated regularly and is available through the Foldseek web server and can be downloaded using the 'databases' command.

**Complex structure prediction.** Four *Sulfitobacter* sp. JL08 protein sequences, identified as CRISPR–Cas type IV-A components by Altae-Tran et al.[14]—Csf1, Csf2, Csf3 and Cas6—were obtained from the plasmid map 'pHS1068 NZ_CP025815 DinG HNH proteins (*Escherichia coli* codon optimized) CRISPR array in pACYCDuet-1 with Lac promoters. gb', released by the authors. Following the reported stoichiometry of the CRISPR–Cas type IV-A core complex[26], we constructed an input file for ColabFold-AlphaFold-Multimer[15] with eight chains: 1xCsf1 + 5xCsf2 + 1xCsf3 + 1xCas6. When examining the structure, we noticed that AlphaFold-Multimer did not predict an interaction between Csf1 and Cas6 and the rest of the complex, so we omitted them and re-predicted the structure: 5xCsf2 + 1xCsf3. Comparing the four sequences of *Sulfitobacter* sp. JL08 to protein nr[27] was performed using the blastp web server (February 2024).

**Runtime evaluation.** Performance was measured on a server with a 1x AMD EPYC 7702 64-core CPU and 1 TB RAM, using a single core. Owing to its high computational demand, the total runtime was extrapolated when measuring US-align on the *Sulfitobacter* sp. JL08 structure against the PDB100, using five samples: avg $= \frac{1}{5} \sum_{i=1}^{5} t(q, \text{sample}_i) \frac{426,347}{1,000}$.

Foldseek-Multimer was run against the full database, without extrapolation. For the MacBook runtime measurements, we used a 13-inch MacBook Pro (M1; 2020; model A2338) with 16 GB RAM.

### Comparison to QSalign on 3DComplexV7

**QSalign pairs.** Starting with the list of 57,953,513 QSalign-compared pairs, high-scoring homomeric pairs (maximum TM score ≥ 0.65) were selected, excluding pairs with a PISA structure. This resulted in 4,524,503 structurally similar unique homomeric pairs, which we denoted 'QSalign pairs'.

**A clustered 3DComplexV7.** The 557,146 chains of 3DComplexV7 were clustered using Foldseek (parameters: `-c 0.99 --min-seq-id 0.9 -e 0.00001`), resulting in 142,957 redundancy-reduced representatives. This procedure took 18 s, using 64 threads.

**Foldseek-MM all-versus-all search of 3DComplexV7.** During this search, all temporary files were kept in memory and 128 cores were used (2 × AMD EPYC 7742). The entire search finished in 10 h and 23 min. Most of the time was spent in the module for matching chains, which took 7 h and 32 min.

**Evaluation of 'Foldseek-MM pairs'.** About 1.7 million pairs of homomeric complexes were detected only by Foldseek-MM as similar. Since running US-align over all pairs is prohibitively slow, we randomly selected 160,252 pairs (around 10% of all pairs) and computed their alignment using US-align. For 2,844 of these (1.8%), US-align reported a TM score < 0.65, which we used as an estimate for the false-positive rate among the full set of novel 'Foldseek-MM pairs'. Around 157,391 pairs (98.2%) were confirmed as matches by US-align and the rest (17 pairs, <0.0001%) were aligned as monomers.

### The BFMD resource

In an effort to generate a large multimer database, we gathered 297,570 multimer predictions, consisting of 597,640 chains from several

community efforts. These were turned into a clustered Foldseek database using the parameters: `-c 0.95 --min-seq-id 1.0 -e 0.00001`, resulting in 51,757 redundancy-reduced representatives. All predictions' accessions are prefixed by the resource name. Multimers extracted from the `ModelArchive`[19], all-versus-all prediction of a set of human genome maintenance proteins `Predictomes`[20], LevyLab atlas of predicted homomers[10], protein–protein prediction from the Human Reference Interactome[28] and the Human Protein Complex Map[29] `HuIntAF2` (ref. 11) and `ProtVar`, predicted multimers as part of an effort to understand missense variance[21]. The BFMD is available through the Foldseek web server and is downloadable as a standalone database using the 'databases' module.

### Tool commands and arguments
Foldseek-MM commit c27a629 (default, using 3Di+AA):

```
foldseek easy-complexsearch query.pdb
target.pdb/targetDB result tmp –threads 1
```
Foldseek-MM-TM commit c27a629 (using tmalign):
```
foldseek easy-complexsearch query.pdb
target.pdb/targetDB result tmp –threads 1
–alignment-type 1
```

Additionally, the flag '`--exhaustive-search 1`' was used for the benchmark of similar pairs and the flag '`--cluster-search 1`' was used when using a clustered db. For database searches, we pre-indexed the database using '`createindex targetDB`' and kept it in memory. We set `--db-load-mode` to 2 in easy-complexsearch, to indicate that the pre-indexed database is already in memory. During database search, Foldseek-Multimer can include or reject monomeric targets in the reference database using the `--monomer-include-mode` parameter. For this study, we set the parameter to reject all monomer matches.

US-align version 20220924:

```
US-align query.pdb target.pdb -mm 1 -ter 0 -mol prot
```

Additionally, the flag '`-fast`' was set for during runtime assessments in Fig. 2b. For speed measurements, we kept the PDB/mmCIF files in memory to avoid input/output-related bottlenecks.

### Reporting summary
Further information on research design is available in the Nature Portfolio Reporting Summary linked to this article.

## Data availability
The benchmarking and 3DComplexV7 data are available via Zenodo at https://doi.org/10.5281/zenodo.13121434 (ref. 30) and the PDB via https://files.wwpdb.org/pub/pdb/data/structures/all/.

## Code availability
Foldseek-Multimer and its web server are GPLv3-licensed free open-source software. The source code and binaries for Foldseek-Multimer can be downloaded at https://github.com/steineggerlab/foldseek/. The analysis scripts are available at https://github.com/steineggerlab/foldseek-multimer-analysis/. The web server is available at https://search.foldseek.com/ and its source code at https://github.com/soedinglab/mmseqs2-app/.

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

## Acknowledgements
We thank H. Kim for helping with designing Fig. 1, Y. Hwan Kim for suggesting the name Foldseek-Multimer and D. Burstein for his feedback on the CRISPR–Cas example. We thank the EMBL-EBI for releasing the 'EMBL-EBI Icon fonts for the life sciences' that are used in the web server. M.S. acknowledges support by the National Research Foundation of Korea (grants 2020M3-A9G7-103933, 2021-R1C1-C102065, 2021-M3A9-I4021220 and RS-2024-00396026), Samsung DS research fund, Creative-Pioneering Researchers Program and AI-Bio Research Grant through Seoul National University. M.M. acknowledges support from the National Research Foundation of Korea (grant RS-2023-00250470). E.D.L. acknowledges support from the European Research Council under the European Union's Horizon 2020 research and innovation program (grant agreement no. 819318).

## Author contributions
W.K., J.S. and M.S. designed the Foldseek-Multimer algorithm. W.K., M.M. and M.S. developed the software. M.M. and C.L.M.G. developed the Foldseek-Multimer web server. E.D.L. and H.S. designed the comparison to QSalign, provided 3DComplexV7 and applied QSalign to it. E.L.K. developed the CRISPR–Cas example. W.K., E.L.K. and M.S. designed the figures, performed the benchmarks and wrote the manuscript, with contributions from all authors.

## Funding

## Competing interests
M.S. acknowledges outside interest in Stylus Medicine. The remaining authors declare no competing interests.

## Additional information

**Correspondence and requests for materials** should be addressed to Emmanuel D. Levy or Martin Steinegger.

# natureportfolio

# Reporting Summary

## Statistics

For all statistical analyses, confirm that the following items are present in the figure legend, table legend, main text, or Methods section.

| n/a | Confirmed | |
|---|---|---|
| ☐ | ☒ | The exact sample size (*n*) for each experimental group/condition, given as a discrete number and unit of measurement |
| ☐ | ☒ | A statement on whether measurements were taken from distinct samples or whether the same sample was measured repeatedly |
| ☐ | ☒ | The statistical test(s) used AND whether they are one- or two-sided<br>*Only common tests should be described solely by name; describe more complex techniques in the Methods section.* |
| ☒ | ☐ | A description of all covariates tested |
| ☒ | ☐ | A description of any assumptions or corrections, such as tests of normality and adjustment for multiple comparisons |
| ☐ | ☒ | A full description of the statistical parameters including central tendency (e.g. means) or other basic estimates (e.g. regression coefficient) AND variation (e.g. standard deviation) or associated estimates of uncertainty (e.g. confidence intervals) |
| ☐ | ☒ | For null hypothesis testing, the test statistic (e.g. *F*, *t*, *r*) with confidence intervals, effect sizes, degrees of freedom and *P* value noted<br>*Give P values as exact values whenever suitable.* |
| ☒ | ☐ | For Bayesian analysis, information on the choice of priors and Markov chain Monte Carlo settings |
| ☒ | ☐ | For hierarchical and complex designs, identification of the appropriate level for tests and full reporting of outcomes |
| ☐ | ☒ | Estimates of effect sizes (e.g. Cohen's *d*, Pearson's *r*), indicating how they were calculated |

*Our web collection on statistics for biologists contains articles on many of the points above.*

## Software and code

Policy information about availability of computer code

| Data collection | Software used in benchmark :<br>- Foldseek(commit c27a629) https://github.com/steineggerlab/foldseek/<br>- USalign (version 20220924) https://zhanggroup.org/US-align/bin/module/USalign.cpp |
|---|---|
| Data analysis | The analysis scripts are publicly available at https://github.com/steineggerlab/foldseek-multimer-analysis.<br>The anaysis data are publicly available at https://doi.org/10.5281/zenodo.13121434. |

For manuscripts utilizing custom algorithms or software that are central to the research but not yet described in published literature, software must be made available to editors and reviewers. We strongly encourage code deposition in a community repository (e.g. GitHub). See the Nature Portfolio guidelines for submitting code & software for further information.

## Data

Policy information about availability of data

All manuscripts must include a data availability statement. This statement should provide the following information, where applicable:

- Accession codes, unique identifiers, or web links for publicly available datasets
- A description of any restrictions on data availability
- For clinical datasets or third party data, please ensure that the statement adheres to our policy

Databases:

## Human research participants

Policy information about studies involving human research participants and Sex and Gender in Research.

| | |
|---|---|
| Reporting on sex and gender | n/a |
| Population characteristics | n/a |
| Recruitment | n/a |
| Ethics oversight | n/a |

Note that full information on the approval of the study protocol must also be provided in the manuscript.

# Field-specific reporting

Please select the one below that is the best fit for your research. If you are not sure, read the appropriate sections before making your selection.

☒ Life sciences ☐ Behavioural & social sciences ☐ Ecological, evolutionary & environmental sciences

For a reference copy of the document with all sections, see nature.com/documents/nr-reporting-summary-flat.pdf

# Life sciences study design

All studies must disclose on these points even when the disclosure is negative.

| | |
|---|---|
| Sample size | The 3DComplex database<br>We downloaded the 3DComplex database version 7 (3DComplexV7DB). This database holds 238,965 structures, consisting of 557,146 chains and was created from the "Biological Units/Assemblies" downloaded from the PDB (Levy et al.).<br><br>Pairwise benchmark<br>Starting with the list of 57,953,513 QSalign-compared 3DComplexV7 structures, pairs of complexes were selected per number of subunits, with that number ranging from 2 to 24. For each size, the criteria for selection were that the TMscore computed by Kpax was greater than 0.8, and that pairs of homomers had less than 80% sequence identity. If more than 100 pairs matched the criteria, only the first 100 were selected, resulting in a total of 931 complex pairs included in the benchmark. These sample sizes are sufficient because we either collected all pairs in the database that matched the similarity criteria (so no sampling) or sampled 100, which is sufficient according to the central limit theorem as we report the difference in average runtimes. We could not perform the measurement on all pairs if there were more than 100, because of USalign's prohibitively slow runtimes.<br><br>PDB100 database<br>First, PDB, containing the asymmetric unit of 207,937 entries, consisting of 1,047,615 chains, was downloaded in November 2023 (Data Availability). Of these, 11,901 entries were associated with more than one structural model (e.g., the NMR experiment 2KOX). In total, 426,347 structural models were associated with the PDB entries. Next, all chains were clustered using Foldseek resulting in 343,785 redundancy reduced representatives. |
| Data exclusions | We extrapolated runtime of US-algin from a random sample of 1000 structures in Fig2b(bottom) and Fig.2c otherwise no data was excluded. |
| Replication | We checked that our method produces deterministic results. We published scripts and data to make the study reproducible. |
| Randomization | No randomization was performed since it was not relevant to this study. Not a comparison across groups - each tool analyzes all data points. |
| Blinding | No blinding was performed as it is not relevant to this study. Not a comparison across groups - each tool analyzes all data points. |

# Reporting for specific materials, systems and methods

We require information from authors about some types of materials, experimental systems and methods used in many studies. Here, indicate whether each material, system or method listed is relevant to your study. If you are not sure if a list item applies to your research, read the appropriate section before selecting a response.

## Materials & experimental systems

| n/a | Involved in the study |
|-----|----------------------|
| ☒ | Antibodies |
| ☒ | Eukaryotic cell lines |
| ☒ | Palaeontology and archaeology |
| ☒ | Animals and other organisms |
| ☒ | Clinical data |
| ☒ | Dual use research of concern |

## Methods

| n/a | Involved in the study |
|-----|----------------------|
| ☒ | ChIP-seq |
| ☒ | Flow cytometry |
| ☒ | MRI-based neuroimaging |

