## [Peer Review File · Nature Methods]

Rapid and Sensitive Protein Complex Alignment with Foldseek-Multimer

Corresponding Author: Professor Martin Steinegger

Version 0:

Decision Letter:

1st Jul 2024

Dear Martin,

Your Brief Communication, "Rapid and Sensitive Protein Complex Alignment with Foldseek-Multimer", has now been seen by 2 reviewers. As you will see from their comments below, although the reviewers find your work of considerable potential interest, they have raised a few concerns. We are interested in the possibility of publishing your paper in Nature Methods, but would like to consider your response to these concerns before we reach a final decision on publication.

We therefore invite you to revise your manuscript to address these concerns.

Link Redacted

We hope to receive your revised paper within 4 weeks. If you cannot send it within this time, please let us know. In this event, we will still be happy to reconsider your paper at a later date so long as nothing similar has been accepted for publication at Nature Methods or published elsewhere.

OPEN SCIENCE REQUIREMENTS

REPORTING SUMMARY AND EDITORIAL POLICY CHECKLISTS

IMAGE INTEGRITY

DATA AVAILABILITY

All novel DNA and RNA sequencing data, protein sequences, genetic polymorphisms, linked genotype and phenotype data, gene expression data, macromolecular structures, and proteomics data must be deposited in a publicly accessible database, and accession codes and associated hyperlinks must be provided in the "Data Availability" section.

CODE AVAILABILITY

Please include a "Code Availability" subsection in the Online Methods which details how your custom code is made available. Only in rare cases (where code is not central to the main conclusions of the paper) is the statement "available

upon request" allowed (and reasons should be specified).

MATERIALS AVAILABILITY

SUPPLEMENTARY PROTOCOL

To help facilitate reproducibility and uptake of your method, we ask you to prepare a step-by-step Supplementary Protocol for the method described in this paper. We [encourage authors to share their step-by-step experimental protocols](https://www.nature.com/nature-research/editorial-policies/reporting-standards#protocols) on a protocol sharing platform of their choice and report the protocol DOI in the reference list. Nature Portfolio's protocols.io is a free-to-use and open resource for protocols; protocols deposited onto protocols.io are citable and can be linked from the published article. More details can found at [protocols.io](https://www.protocols.io/help/publish-articles).

ORCID

Nature Methods is committed to improving transparency in authorship. As part of our efforts in this direction, we are now requesting that all authors identified as 'corresponding author' on published papers create and link their Open Researcher and Contributor Identifier (ORCID) with their account on the Manuscript Tracking System (MTS), prior to acceptance. This applies to primary research papers only. ORCID helps the scientific community achieve unambiguous attribution of all scholarly contributions. You can create and link your ORCID from the home page of the MTS by clicking on 'Modify my Springer Nature account'. For more information please visit please visit www.springernature.com/orcid.

Sincerely,
Arunima

Arunima Singh, Ph.D.
Senior Editor
Nature Methods

Reviewers' Comments:

Reviewer #1 (Remarks to the Author):

This paper describes an extension to the widely used FoldSeek algorithm. The paper is well written, and the results are convincing - the performance is on par with the state-of-the-art method. Still, the algorithm is orders of magnitude faster, necessary for analyzing the many large structures coming out from Cryo-EM and recent predictions. I have tested the program, and it is easy to run. Lastly, the algorithm is licensed under GPLv3, guaranteeing it will be widely adopted in the scientific community.

I only have one minor comment about using the method.

When you run the program in the "easy-cluster" modelling, it does not cluster on the complex level. Instead, it clusters at the single chain level, which requires an external clustering algorithm.

/Arne

Reviewer #1 (Remarks on code availability):

Code is available - works and licensed freely

Reviewer #2 (Remarks to the Author):

In this work, Kim et al. present Foldseek-Multimer, a novel tool for rapid alignment of protein complexes. The authors introduce (1) the concept of superposition vector clustering to map chains between complexes, (2) an extension of Foldseek searches on pre-clustered databases to further speed up the pre-filter and alignment steps. Overall, the method is of high relevance and impact and the benchmarking results are convincing as a much faster drop-in replacement to other commonly used methods in their typical use-cases for this task.

Comments:

In a number of places, sequence identity is used where sequence similarity would be more appropriate given the claim of "owed to its ability to detect similar complex structures, even with little shared sequence similarity". E.g, the statement "Intrigued by how evolutionarily distant this system was to known proteins (<65% sequence identity to any entry in the nr database (15)),..." is not so impressive as 65% sequence identity is quite high. In general, this added ability to detect many more complex pairs due to the reduced time complexity is very exciting but under-described in its potential to detect remote homology complexes. A supplementary analysis of sequence similarity of the 1.6 million Foldseek-MM-detected homomeric pairs would drive this point home. Related, the effect of prediction error (e.g pIDDT) on chain mapping is not described, to better support the claim of "sensitive and suitable for metagenomic studies of complexes with low sequence identity to others".

The superposition vector clustering is a novel approach to chain mapping, but it is difficult to follow its motivations as it's written. For example, a sentence like: "If chain A in complex Q is positioned and oriented relative to chain B in complex Q in a similar way to how chain A in complex T is positioned and oriented relative to chain B in complex T, then the superpositions of QA-TA and QB-TB will be similar. This similarity in superposition indicates that the two chain-to-chain alignments are compatible with the same overall alignment of the two protein complexes." could help readers understand the underlying concept. A supplementary figure would also be helpful to expand on Figure 1c-d and describe the whole process including the cluster rescuing.

If I understand correctly, the clustering algorithm does not have to assign chains for every chain in the query complex, i.e it's a "local" chain mapping rather than a global one. If so, I would also expect a figure/result about the resulting coverage of the complex assignments found. In general, a supplementary figure comparing the exact chain mappings found by FoldSeek-MM and US-align on the same benchmark sets used would be helpful to show the robustness of the chain mapping algorithm.

While runtimes are shown across different numbers of chains, it would also be interesting to see the TMscore as a function of chain count (see e.g Fig 2a-b in <https://doi.org/10.1038/s41592-022-01585-1>). Some words on the difference in results between homomeric and heteromeric complexes would also be interesting. It's unclear why only homomeric pairs were used in the 3DComplexV7 benchmark.

There are a few hard-coded parameters in the algorithm: $CV < 0.1$, epsilon delta, the threshold differentiating core and non-core points. Could some explanation be given on how these were chosen? Also, why are the parameters different for PDB100 and 3DComplexV7 clustering, and would the clustering parameters affect the multimer search results?

Reviewer #2 (Remarks on code availability):

The code is usable, stable and "fit for purpose".

Version 1:

Decision Letter:

Our ref: NMETH-BC56426A

30th Sep 2024

Dear Martin,

Thank you for submitting your revised manuscript "Rapid and Sensitive Protein Complex Alignment with Foldseek-Multimer" (NMETH-BC56426A). It has now been seen by the original referees and their comments are below. The reviewers find that the paper has improved in revision, and therefore we'll be happy in principle to publish it in Nature Methods, pending minor revisions to satisfy the referees' final requests and to comply with our editorial and formatting guidelines.

We are now performing detailed checks on your paper and will send you a checklist detailing our editorial and formatting requirements within two weeks or so. Please do not upload the final materials and make any revisions until you receive this

additional information from us.

TRANSPARENT PEER REVIEW

ORCID

Sincerely,
Arunima

Arunima Singh, Ph.D.
Senior Editor
Nature Methods

Reviewer #1 (Remarks to the Author):

I am happy with the authors respons and I look forward to using the complex search once it is carefully benchmarked

Open Access This Peer Review File is licensed under a Creative Commons Attribution 4.0 International License, which permits use, sharing, adaptation, distribution and reproduction in any medium or format, as long as you give appropriate credit to the original author(s) and the source, provide a link to the Creative Commons license, and indicate if changes were

made.

Dear Arunima,

Thank you for handling our manuscript so promptly. We also want to express our sincere gratitude to the reviewers for their comments.

We addressed all points raised by them in our point-by-point response below. Briefly, we have added several supplementary figures that explain our cluster rescuing procedure (Supp. Fig. 1), disentangle Foldseek-Multimer's performance on different chain-counts (Supp. Fig. 2), show Foldseek-Multimer's performance when used in conjunction with predicted structures (Supp. Fig. 4), provide deeper analysis of Foldseek-Multimer's performance on the 3DComplexV7 database (Supp. Fig. 5) and the effect of hard-coded parameters (Supp. Table 1) and have overall improved the clarity of the manuscript.

Since the submission of the manuscript, we have added a new resource, the BFMD database, which gathers predictions from several large-scale community efforts for multimer prediction and has been made available for local and webserver use. Additionally, we are proud to report that Foldseek-Multimer is being rapidly adopted by the community, with its webserver having already processed over two thousand user-submissions.

Best regards,
Emmanuel and Martin

Reviewer #1:

This paper describes an extension to the widely used FoldSeek algorithm. The paper is well written, and the results are convincing - the performance is on par with the state-of-the-art method. Still, the algorithm is orders of magnitude faster, necessary for analyzing the many large structures coming out from Cryo-EM and recent predictions. I have tested the program, and it is easy to run. Lastly, the algorithm is licensed under GPLv3, guaranteeing it will be widely adopted in the scientific community.

Thank you very much for the positive feedback.

I only have one minor comment about using the method. When you run the program in the "easy-cluster" modelling, it does not cluster on the complex level. Instead, it clusters at the single chain level, which requires an external clustering algorithm.

We are currently in the early development stages of a complex-level clustering algorithm. We do this as a separate project since a complex-level clustering algorithm has additional requirements, like resolving the $O(N^2)$ complexity with an adjusted linclust and the non-transitivity of partially-matching clusters. The transitivity is an important issue, since Foldseek-Multimer can detect similarities in which only some of the chains match (partial match). Thus, its set of detected similarities is not necessarily transitive (e.g., if complex A matches complex B due to similarities between chains A1, A2 and B1, B2 and complex B matches complex C due to similarities between B2, B3 and C1, C2 then complexes A and C may not be similar and should not be clustered).

As part of our commitment to open-science, the in-development version of the new algorithm is already available at <https://github.com/rachelse/foldseek>, and will be made available within the main Foldseek repository as soon as we are confident about its performance with thorough benchmarks.

Reviewer #2 (Remarks to the Author):

In this work, Kim et al. present Foldseek-Multimer, a novel tool for rapid alignment of protein complexes. The authors introduce (1) the concept of superposition vector clustering to map chains between complexes, (2) an extension of Foldseek searches on pre-clustered databases to further speed up the pre-filter and alignment steps. Overall, the method is of high relevance and impact and the benchmarking results are convincing as a much faster drop-in replacement to other commonly used methods in their typical use-cases for this task.

Thank you very much for the positive feedback.

Comments:

In a number of places, sequence identity is used where sequence similarity would be more appropriate given the claim of "owed to its ability to detect similar complex structures, even with little shared sequence similarity". E.g, the statement "Intrigued

by how evolutionarily distant this system was to known proteins (<65% sequence identity to any entry in the nr database (15)),...” is not so impressive as 65% sequence identity is quite high.

We agree and addressed all three occurrences of this issue in the revised manuscript:

1) Concerning the CRISPR-Cas ribonuclease,

- We think the key point is that it is the first discovered CRISPR-Cas type IV-A system with a specified interference mechanism, so we removed the part about sequence identity and now write:

“Recently, Altae-Tran et al. (2023) discovered the first CRISPR-Cas type IV-A system with a specified interference mechanism in an environmental sample of *Sulfitobacter* sp. JL08. Intrigued by their finding, we predicted a part of its ribonucleoprotein complex structure using ColabFold-AlphaFold-Multimer.”

- When reporting the search results, we added information about sequence similarity: “...despite low sequence similarity (11.1-19.8% sequence identity and 19-33.3% sequence similarity using the BLOSUM62 substitution matrix) between the six subunit pairs...”

2) In the introduction, we replaced the word “identity” with “similarity” in the following:

“sensitive and suitable for metagenomic studies of complexes with low sequence **similarity** to others”

In general, this added ability to detect many more complex pairs due to the reduced time complexity is very exciting but under-described in its potential to detect remote homology complexes. A supplementary analysis of sequence similarity of the 1.6 million Foldseek-MM-detected homomeric pairs would drive this point home.

We thank the reviewer for this suggestion and added **Supp. Fig. 5**, where we measured the median sequence identity and similarity for each of the 1.7 million homomeric pairs exclusively detected by Foldseek-MM. Most complex pairs (85%) were highly diverged, having less than 50% sequence similarity. Furthermore, these results demonstrate Foldseek-Multimer's ability to detect structural similarity well below the twilight zone, as 87% of the pairs had less than 30% sequence identity between the complexes. We now refer to Supp. Fig. 5 from the revised main text.

Related, the effect of prediction error (e.g pLDDT) on chain mapping is not described, to better support the claim of “sensitive and suitable for metagenomic studies of complexes with low sequence identity to others”.

Thank you for this comment, we acknowledge this is an important issue, but it is difficult to answer reliably since we have no ground truth set with variable pTM/pLDDT values. In an attempt to test the effect of structure prediction quality on the ability of Foldseek-Multimer to find and score matches, we prepared a setup where we could try to compare Foldseek-Multimer's results on predicted query structures of varying quality to those for true structures. To that end, we used a part of the Fig. 2a dataset, where the queries were PDB structures, which have been determined using gold-standard techniques and can therefore be considered as "true" structures. Specifically, we used ColabFold-AlphaFold2 v1.5.5 to predict the structures of dimer queries from this dataset, setting its parameters to produce structures of various qualities (--use-dropout --num-seeds 10 --num-recycle 6). We then

used Foldseek-Multimer to search each predicted query complex against the PDB100 database, recorded its top hit and score, and compared it to the top hit and score recorded when the true structure was used as query. We found that higher quality predictions (using the pTM metric, which is recommended for complexes more than pLDDT) tend to result in better FS-MM TM-scores. At the same time, despite the variance in prediction quality, all but two (98.5%) predicted structures had a FS-MM top hit with a TM-score > 0.65. Of these, most (87%) had a very small FS-MM TM-score difference (< 0.05) when querying with the predicted and the true structure. Predicted structures that were less similar to the true one (lower pTM) were more likely to be matched by FS-MM with a different top hit or have a worse score. This association is statistically significant (Chi-square P-value < 0.00001). We present these findings in **Supp. Fig. 4** of the revised manuscript.

The superposition vector clustering is a novel approach to chain mapping, but it is difficult to follow its motivations as it's written. For example, a sentence like: "If chain A in complex Q is positioned and oriented relative to chain B in complex Q in a similar way to how chain A in complex T is positioned and oriented relative to chain B in complex T, then the superpositions of QA-TA and QB-TB will be similar. This similarity in superposition indicates that the two chain-to-chain alignments are compatible with the same overall alignment of the two protein complexes." could help readers understand the underlying concept.

Thank you. We agree that this is a key point in the Foldseek-Multimer algorithm that deserves to be presented clearly. We therefore extended the "Algorithm: Overview" section, incorporating the example suggested by the reviewer. In addition, we made minor modifications to Figure 1, which we hope will contribute to its clarity and refer to the revised "Algorithm: Overview" section from Figure 1 legend. We now write:

"Foldseek-Multimer examines all possible chain-to-chain pairings between the compared complexes, using Foldseek (Fig. 1b). It then uses the fact that a structural alignment between two complexes, **Q and T**, implies a superposition: a set of rotations and translations, which minimize the sum of squared distances between their aligned residue pairs (19). **For simplicity, assume Q and T to be two structurally similar dimers, consisting of the chains Q_A, Q_B and T_A, T_B, where Q_A is similar to T_A and Q_B is similar to T_B. The physical meaning of the complex-level structural similarity is that Q_A is positioned and oriented relative to Q_B within Q in the same way that T_A is positioned and oriented relative to T_B within T. Thus, the same superposition, i.e., the same set of rotations and translations, would minimize the distance between Q_A and T_A as well as the distance between Q_B and T_B. In other words, all individual chain-to-chain superpositions (e.g., the one between Q_A and T_A) are equal to one another and to the complex-to-complex superposition. Therefore, a set of chain-to-chain alignments is compatible and can define a complex-to-complex alignment, only if all chain-to-chain superpositions computed from that set are equal.** Foldseek-Multimer therefore computes for each chain-to-chain alignment a vector, representing its superposition (Fig. 1c). Next, it uses DBSCAN (20) for clustering these vectors to identify compatible sets of chain-to-chain alignments, which share the same superposition and define valid complex alignments (Fig. 1d). Once complex alignments are identified, Foldseek-Multimer computes their TM-score (21) and reports them (Fig. 1e)."

A supplementary figure would also be helpful to expand on Figure 1c-d and describe the whole process including the cluster rescuing.

Thank you for this comment, it made it clear to us that the description of the iterative DBSCAN was lacking. We revised the section "Algorithm: Chain-to-chain clustering" to provide the full details of the procedure, including the subsections: "The DBSCAN iteration", "Cluster validity and rescuing by Nearest Neighbors", "Iterativity" and "Discovered clusters". We also added **Supp. Fig. 1**, extending panels 1c-d, as suggested by the reviewer. In the revised version, we refer to this figure both from the legend of Fig. 1 and from the revised section "Algorithm: Chain-to-chain clustering".

If I understand correctly, the clustering algorithm does not have to assign chains for every chain in the query complex, i.e it's a "local" chain mapping rather than a global one. If so, I would also expect a figure/result about the resulting coverage of the complex assignments found. In general, a supplementary figure comparing the exact chain mappings found by FoldSeek-MM and US-align on the same benchmark sets used would be helpful to show the robustness of the chain mapping algorithm.

As the reviewer suggested, we examined US-align and Foldseek-Multimer's chain-to-chain assignments in the dataset of 931 structurally-similar complex pairs used for Fig. 2a. In case Foldseek-Multimer finds several equal highest-scoring solutions (chain assignments) it reports all of them, while US-align reports only one solution. In 923 of the cases (99.1%), the chain assignment reported by US-align, which included all chains, was found amongst the highest-scoring solutions reported by Foldseek-MM and Foldseek-MM-TM. In four cases (0.4%) Foldseek-MM and Foldseek-MM-TM reported a different chain assignment than US-align but covered all chains. In two cases (0.2%) Foldseek-MM and Foldseek-MM-TM did not report any alignment and in two cases (0.2%) Foldseek-MM and Foldseek-MM-TM missed one or two chains, which were reported by US-align. In one case (0.1%) Foldseek-MM-TM reported a different chain assignment and Foldseek-MM missed one chain, relative to the US-align solution. In the revised version we added to the Main:

"...Foldseek-Multimer computed highly correlated TM-scores to those of US-align (Fig. 2a) and produced the same chain pairing in >99% of the cases (Data Availability)."

While runtimes are shown across different numbers of chains, it would also be interesting to see the TMscore as a function of chain count (see e.g Fig 2a-b in <https://doi.org/10.1038/s41592-022-01585-1>).

In the revised version, we modified **Supp. Fig. 2** to provide more details about the analysis of Fig. 2a. Supp. Fig. 2 now shows the TM-score correlation as a function of the normalization method and number of chains. Specifically, Fig. 2a presents the correlation between query-length normalized TM-scores computed by either Foldseek-Multimer or US-align for 931 pairs of structurally similar complexes. As per the reviewer's suggestion, this computation was repeated separately for complexes of the same number of chains using either query- or target-length normalization if there were at least 30 complexes of that size (i.e., separately for complexes with 2, 3, 4, 5, 6, 8, 10, 12, 14, and 24 chains). The correlation was high in all cases (Pearson's r : 0.88-1.0). We therefore include in Supp. Fig. 2 eight additional panels, showing these results for a selected number of chains.

Some words on the difference in results between homomeric and heteromeric complexes would also be interesting. It's unclear why only homomeric pairs were used in the 3DComplexV7 benchmark.

We initially analyzed both homo- and heteromeric pairs for the first bioRxiv submission (<https://www.biorxiv.org/content/10.1101/2024.04.14.589414v1>) based on 3DComplex data. However, we realized shortly thereafter that heteromers make it sometimes difficult to interpret similarities and differences in TM-score. In particular, when considering heteromeric complexes containing subunits with different sizes, two complexes may yield a similar TM-score (>0.65) because a large subunit matches while a small one does not. This issue and others do not exist for homo-oligomers, so we decided to focus only on the homo-oligomers for the second bioRxiv version and this manuscript submission.

There are a few hard-coded parameters in the algorithm: $CV < 0.1$, epsilon delta, the threshold differentiating core and non-core points. Could some explanation be given on how these were chosen?

Thank you for this comment. Our algorithm has four hard-coded parameters: CV , ϵ , ϵ -delta and min-points. Min-points has to be set to 2 to allow for dimer clustering. Therefore, we evaluated the effect of the other three parameters by testing seven alternative parameter sets (**Supp. Table 1**). Here, we repeated the analysis conducted for Fig. 2a with the alternative parameter sets and report the Pearson correlation and regression coefficients with the US-align scores and runtimes. Although we did not observe an improvement in Foldseek-Multimer's sensitivity, changes to ϵ and ϵ -delta resulted in worse runtime. Therefore, we decided to retain the hard-coded settings.

Also, why are the parameters different for PDB100 and 3DComplexV7 clustering, and would the clustering parameters affect the multimer search results?

PDB100 was introduced first as part of the ColabFold structural template search and the parameters were set such that no template would be lost, so only identical (sub-)sequences of similar length are clustered together (`--min-seq-id 1 -c 0.95`), producing a redundancy-reduced set of representatives. In contrast, 3DComplexV7 was first clustered as part of this project. It associates representatives with the complexes and chains in their cluster, creating a cluster db and not just a set of representatives. Using this association, *cluster search* reaches any complex and its chains from the representative. Therefore, setting the `--min-seq-id` parameter to 0.95 does not risk losing matching complexes; it only reduces the number of representatives.

We have also converted the redundancy-reduced PDB100 to a cluster database, thereby enabling searches against all PDB complexes. Reclustering the PDB with `--min-seq-id 0.95` or lower would have the potential to improve the speed of Foldseek-Multimer searches but not sensitivity. However, we opted not to change clustering parameters so that we do not maintain multiple highly similar databases for different applications.

- Please see the author guidance document for a list of figures and figure legends that require additional information.

We updated the figure.

- Please mark this field as "confirmed", as statistical tests like chi-square test, etc. have been referenced in the manuscript.

Confirmed.

- Please mark this field as "confirmed", as p values derived from statistical tests have been referenced in the manuscript.

Confirmed.

- Please provide the github web-link (<https://github.com/soedinglab/mmseqs2-app>) for the codes , here in the reporting summary as well.

We have added mmseqs2-app to the code availability section.

- Please ensure the following softwares/packages/tools/algorithms are mentioned in the manuscript as well, since they have been listed in the reporting summary: Python, etc.

We have updated the reporting summary accordingly.

- Please ensure all the data collection/data analysis softwares/tools/algorithms/packages mentioned in the manuscript are also listed in the reporting summary (with version numbers): DBSCAN algorithm, etc.

DBSCAN was not included as we implemented a customized version adapted for protein structure applications; it is described in the Methods section.

- Please ensure that information on datasets used/generated is provided under the 'data' section of the reporting summary, as the information is not relevant to this field.

We have separated the analysis and data sections accordingly.

- Please provide a complete data availability statement in the manuscript under 'data availability' section, as provided here in the reporting summary. Please mention all the databases/datasets used in the study along with appropriately accessible links/accession-codes in the manuscript under the "Data availability" section as well as in this reporting summary: EMBL-EBI database, etc.

The reporting summary is now aligned with the data availability section.